# THE USE OF OPEN SOURCE BOARDS FOR DATA COLLECTION AND MACHINE LEARNING IN REMOTE DEPLOYMENTS

## ABSTRACT

Machine learning is being adopted in many walks of life to solve various problems. This is being driven by development of robust machine learning algorithms, availability of large datasets and low cost computation resources. Some machine learning applications require deployment of devices off-the-grid for data collection and real time monitoring. Such applications require development of systems that can operate autonomously during their deployment. Advancement in technology has seen development of low-cost and low-power open-source microcontrollers and single board computers. These boards can be interfaced with a wide array of sensors and can perform computation processes. The boards are finding wide applications in data collection and machine learning initiatives. This paper will describe how the boards are leveraged for off-grid deployments.

## 1 INTRODUCTION

Machine learning is a discipline that comprises a wide range of algorithms and modeling tools used for diverse data processing tasks. The main goal of machine learning is to recognise patterns in data using computers and give informed insights on how to solve problems (Carleo et al., 2019). Machine learning is one of the most rapidly progressing technical fields. This progress can be attributed to the development of new robust learning algorithms, availability of large volumes of data and low-cost computation resources. Machine learning has been adopted in many fields such as computer vision, pattern recognition, speech recognition, engineering, finance, sciences and healthcare (El Naqa & Murphy, 2015).

Data plays a central role in machine learning. The basic concept of machine learning algorithms is building models that learn from input (training) data and make predictions using new data based on the learnt experience (Lotfian et al., 2021). The development of machine learning is seeing emergence of new applications that lack enough labelled data. Most modern machine learning techniques require a large amount of labelled data (Roh et al., 2019). This makes data collection to be an important step in any machine learning task. Additionally, most of the available datasets are from developed countries and some of these are proprietary. These data may not be adequate to reflect all scenarios due to ecological and geographical differences across the world. This calls for localised data collection initiatives to build datasets that will be appropriate for localised problems solving (Ooko et al., 2021).

One source of data for machine learning applications is sensors. Sensors are devices that detect changes in their surroundings and convert it to an electrical signal. A microprocessor interfaced with the sensor processes the output signal of the sensor and gives an output that corresponds to a set of measures (Javaid et al., 2021). Some applications require deployment of sensors in remote regions that are far from the grid. A good example is ecological data collection using sensors such as acoustic sensors and camera traps. For such off-grid deployments, several issues arise. They include: source of power, storage, on-board data processing, cost of hardware, communication and degree of autonomy for long term deployments (Gibb et al., 2019).

Advancement in technology has seen development of low-cost open-source single board computers and microcontrollers. These boards consume low power, can be interfaced with a wide array of sensors and have relatively sufficient processing capabilities. The boards are finding applications in

off-grid data collection and machine learning tasks. Majority of microcontrollers lack an operating system (OS). These microcontrollers are programmed directly using binary/assembly code or binary code compiled from C-style languages. Single boards like the Raspberry Pi, the BeagleBone and the Jetson Nano have operating systems and can be operated like general purpose desktops (Fitzpatrick et al., 2020; Güven et al., 2017). With TinyML technology algorithms and software that can run on the resource constrained open-access boards have been developed. In this paper we will describe how open-access boards are used for off-grid data collection and machine learning tasks.

## 2 OPEN ACCESS BOARDS

According to Tucson Amateur Packet Radio (TAPR) Open Hardware Licence, 'Open Hardware is a thing - a physical artefact, either electrical or mechanical - whose design information is available to, and usable by, the public in a way that allows anyone to make, modify, distribute, and use that thing' (TAPR.org). Over the years, several low-cost low-power open-source hardware projects have emerged leading to development of standalone systems that can perform computation processes without additional hardware. These hardware include microcontrollers like the Arduino and single board computers like the Raspberry Pi and they are finding applications in diverse fields from industries, homes, education, health and research (Güven et al., 2017). The microcontrollers and single board computers form the heart of systems designed for data collection and processing.

Manual data collection in outdoor field research can be time consuming, labour intensive, costly and irregular. However, using open-source hardware, the process can be automated leading to efficient data collection and even enable real time monitoring (Daniel K & Peter J, 2012). Open source hardware boards with high processing capabilities and interfaces to connect with a wide array of sensors exist in the market. These features make them well suited for data collection and real time monitoring.

The use of open-source hardware in remote data collection and real time monitoring is faced with various constraints that include: (1) source of power; (2) processing capabilities; (3) storage; (4) and communication requirements. The following subsections discuss how these hurdles are overcome while using open-access hardware for off-grid data collection and real time monitoring.

### 2.1 POWERING OPEN SOURCE BOARDS OFF-THE-GRID

For off-grid sensors deployment, alternative methods of powering the sensors need to be devised. Generally, the sensors are powered using batteries due to the intermittent nature of alternative sources like solar and wind (Prauzek et al., 2018; 2016). Despite their ability to provide an uninterruptible power source, batteries are prone to depletion of charge after a given period of usage. By coupling the batteries with other sources of energy such as solar and wind, we can harvest energy from the surroundings. Open source boards have relatively low power requirements in the range of milliwatts to a few watts and they can be powered from DC sources like batteries. A photovoltaic and battery system can be used to power the open-source boards for long term deployments. The boards can also be programmed to have flexible operation schedules saving on energy requirements of the systems.

### 2.2 DATA COLLECTION, STORAGE AND PROCESSING

Most open-source boards can be interfaced with a wide array of sensors like temperature sensors, cameras, microphones and ultrasonic sensors. The sensors are interfaced using the general purpose input output (GPIO) pins, CSI ports or USB ports that come with the boards. Some boards like the Arduino also have on-board sensors. The sensors enable the boards to interact with the outside world and are used for data collection. The data collected by the boards can either be transmitted to a central hub or the cloud for storage or stored in storages such as SD cards and USB flash disks.

Deployments of sensors lead to collection of large volumes of data. The data needs to be sorted, cleaned and labelled before it can be used for machine learning applications. These tasks are, however, difficult and time consuming. Onboard processing of data can be used to filter out unnecessary data and only save the appropriate data. This can be achieved by adding filters in the data acquisition pipeline. The filters are achieved by setting thresholds of certain parameters of the data that

is being collected. An example is the energy of a signal in sound data collection to detect acoustic activities. Filtering the data stream greatly reduces the work needed to clean the data and save on storage. There is a tradeoff however when it comes to onboard processing of data since the hardware consumes more power.

Another way that preprocessing of data can be achieved is by loading the sensors with binary classifiers to filter out unwanted data from the data stream. The binary classifiers need to be pre-trained on a dataset comprising the data of interest and the expected unwanted data. The classifiers can then be added to the data collection pipeline where they will filter out unwanted data and allow only the data of interest to proceed to the next steps.

## 2.3 Machine learning at the edge

To extract information from the data collected by edge devices, computationally expensive algorithms are needed. These algorithms are not compatible with the resources constrained edge devices (Verhelst & Murmann, 2020). The go to solution to this constraint is offloading the data to external computing systems for further processing. This in turn introduces latency, communication requirements and privacy concerns. To solve these issues, efforts are being made to bring the computation tasks to the edge devices and transmit only the necessary data (Murshed et al., 2021). Edge computing involves performing computations close to data sources (Butler, 2017; Satyanarayanan, 2017).

Microcontrollers are resource constrained devices. Conventional machine learning algorithms require a lot of processing power and memory (Low et al., 2012). This makes them unsuitable for deployment on microcontrollers. The machine learning models need to be scaled down to run on microcontrollers. This leads to the concept of TinyML. TinyML technology entails hardware, algorithms and software that are capable of performing data analytics on devices with low processing capabilities. Some commonly used TinyML models include neural networks (NN), support-vector machines (SVM), decision trees, k-nearest neighbours (KNN) and linear regression among many others (Ooko et al., 2021).

Single board computers like the Raspberry Pi possess sufficient computational power to perform machine learning at the edge. The Raspberry Pi is one of the most adopted hardware platforms for ML at edge (Daher et al., 2020). Numerous machine learning tasks have been carried out on it ranging from simple algorithms such as logistic regression, SVM and decision tree to the complex convolutional neural networks (CNNs). Models can be trained on more powerful computers in order to take advantage of their processing capabilities in working on large datasets within a short time. Once the models have been trained and optimised, they are saved and then loaded onto the Raspberry Pi (Curtin & Matthews, 2019). The Raspberry Pi can then be deployed with the necessary sensors in the field for remote monitoring.

Edge computing with the open-source board in remote deployments has the following advantages:

(i) Real time monitoring is possible with minimal latency (ii)] Predictions results can be saved instead of the data reducing the required storage capacity

(iii) It is easier and faster to transmit prediction results compared to the data in many instances

(iv) Prediction results can be used to label collected data

(v) In areas without GSM/ internet connectivity, Low-Power Wide-Area Network (LPWAN) infrastructures can be used to transmit prediction results since they require smaller bandwidth.

(vi) Saves on resources needed to acquire cloud services

## 2.4 Communication

Connectivity of sensor nodes is important especially when real time monitoring is needed. Some open-source boards have onboard wireless communication modules and those that lack can be interfaced with external communication modules. These modules are used to send data, machine learning predictions and the status of the sensor to a central device or the cloud. The boards can be interfaced with GPS/ GPRS modules, Wi-Fi or LPWAN modules like LoRa in places without connectivity for communication purposes.

## 3 USE CASES

This section outlines how different open source hardware are used for data collection and machine learning tasks.

### 3.1 THE RASPBERRY PI

The Raspberry Pi is a low cost credit-card sized single board computer that runs on a Debian based operating system (OS). The board has the ability to do almost everything one would do using a desktop computer (Raspberry-Pi-Foundation, 2015; Hentschel et al., 2016). The Raspberry has been used in numerous off-the-grid data collection and machine learning initiatives. The following are example uses.

#### 3.1.1 DSAIL BIOACOUSTICS SYSTEM

The DSAIL Bioacoustics System is a Raspberry Pi based acoustic sensor developed at the Centre of Data Science and Artificial Intelligence (DSAIL). The acoustic sensor was designed for passive acoustic monitoring (PAM) of ecosystems. The system is powered using a solar panel and a lithium battery pack through a power supply board, the DSAIL Power Management Board. The DSAIL Power Management Board was designed to enhance the autonomy of the system during deployment. The board enables the Raspberry Pi / the user to: (1) monitor the status of the battery; (2) schedule operating windows; (3) safely shutdown the system in event the battery is drained or end of a scheduled operating window; (4) and wake up the system once the battery has been recharged or at the start of an operating window (Kiarie et al., 2022).

The DSAIL Bioacoustics System has been used to collect over a hundred-hours-long sound recordings at the Dedan Kimathi University of Technology Conservancy. Using the power supply board, we are able to schedule the sensor to operate from 5 am to 11 am when birds are most active and shutdown the rest of the day. The battery gets charged by a solar panel and the system wakes up the following day at 5 am (Kiarie & wa MAINA, 2021). Using the data collected by the sensor, machine learning algorithms to perform automatic acoustic classification of birds will be developed and loaded onto the sensor's Raspberry Pi. The system will then be deployed in ecosystems of interest for acoustic monitoring.

#### 3.1.2 DSAIL CAMERA TRAP

The DSAIL Camera Trap is a Raspberry Pi based camera trap designed for wildlife image data collection. The system is also powered using a solar panel and a lithium battery pack through the DSAIL Power Management Board. The system has been deployed at the Dedan Kimathi University of Technology Conservancy for data collection. Over 8,000 images of wild animals have been collected and labelled. The camera trap is set to operate during two time windows in the day. It operates from 6.00 am to 11.00 am then shut down and wakes up at 2.00 pm to 7.00 pm (Mugambi et al., 2015; Mugambi et al.). The data collected by the system will be used to develop machine learning algorithms to classify wild animals from their images for remote monitoring.

#### 3.1.3 SOLO

Solo is a Raspberry Pi based low cost open-source customizable audio recorder designed for bioacoustics research. The system comprises a Raspberry Pi, a PiFace clock module and Cirrus Logic card that can be interfaced with a wide range of external microphones. It can be powered using any 5 V supply. A software to run the system has been developed and is available to the public. The SD card used with the Raspberry Pi is used as the storage for the audio recordings.

SOLO has been deployed at several ecosystems for data collection. Approximately 52,381 hours of audio has been collected using the system. 5 systems have been deployed in the Ebo Forest, southwest Cameroon and they have recorded 600 hours of audio. 10 more systems have been deployed in Central Scotland and Central England and 10,383 hours of audio have been collected. 35 systems were also used to record 41,398 hours of audio in Central Scotland and Central England. With a maximum sampling rate of 192 kHz, SOLO has also been used to make ultrasound recordings of bats (Whytock & Christie, 2017).

### 3.1.4 WIRELESS ACOUSTIC SENSOR SYSTEM (WASS)

WASS is based on a Wireless Acoustic Sensor Network (WASN) that is used to record and transmit the audio samples combined with a classification framework for automated evaluation. WASS was developed for acoustic monitoring of birds. It is made of recording nodes, a gateway node, a backend for data processing and frontend to display results to the end user. The recording nodes are Raspberry Pi based recording systems that are powered via StormPi3 extension board that allows for a wide array of power supplies. The system uses an omnidirectional Rhode VideoMicro and Sabrent AU-MMSA USB soundcard for recording.

The audio recorded by the system is transferred to the cloud through a gateway where data processing and classification is done. The results of the backend are displayed to the user using a frontend that is implemented using asynchronous AJAX calls that is supported by most of the modern browsers even on mobile devices (Brüggemann et al., 2021).

### 3.1.5 A RASPBERRY PI BASED TIGER MONITORING SYSTEM

In order to achieve real time monitoring of Tigers, researchers in Malaysia have developed a wireless image transmission system using the Raspberry Pi. The system is designed to capture, process and transmit images wirelessly to stations of authority. The system is equipped with a Raspberry Pi, a camera and a motion sensor. The motion sensor triggers the system to take an image once there is motion around the camera trap and stores it in an SD card. The captured image is then transmitted over Wi-Fi to a host computer using SAMBA server for further processing. This way the authorities are able to closely monitor tigers and take less time to act when needed (Ahmad et al., 2018).

## 3.2 THE MULTITECH MDOT

The MultiTech mDot is an Arm® Mbed™ programmable, low-power RF module. It provides long-range, low bit rate M2M data connectivity to sensors, industrial equipment and remote appliances. The mDot has an on-board LoRa module that acts as a transceiver (MultiTech). At DSAIL, an mDot based Water Level Data Acquisition System has been developed. The system comprises a Multitech mDot microcontroller, a DC power supply circuit, a US-100 precision range finder ultrasonic sensor and a battery voltage sensor circuit. It is powered using a 3.7 V, 6600 mAh lithium-ion battery and a 9 V 330 mA. The system uses the ultrasonic sensor to determine the level of water in a river channel and transmits the values to the Things Stack Network (TTN) from where it is transferred to the cloud for storage and processing. This data is used to monitor rivers and make inferences on the status of ecosystems (water catchments). The system also transmits the battery voltage for the system's power analysis (Kabi & Maina, 2021; Kabi & wa MAINA).

## 3.3 THE AUDIOMOTH

The AudioMoth is an open-source, low-cost, small-sized and low-energy acoustic detector that is built around an ARM Cortex-M4F microcontroller. It was developed from a research project in Southampton University aimed at monitoring anthropogenic disturbances in the tropical forests of Belize (Open-Acoustics-Devices; Hill et al., 2018). It uses a low-power microcontroller and a microelectromechanical systems (MEMS) microphone to record and analyse sound data.

The AudioMoth is powered using three lithium AA-cell batteries. Its on-board processing capabilities enhance efficient use of the SD card used as its storage [33]. The board has been used for acoustic data collection in ecosystems. One example is its deployment to listen to the sound of shotguns in Belizean tropical forests to identify poaching hotspots. In addition to recording sounds of threats, the AudioMoth collects sounds of wildlife. The data is available to develop machine learning algorithms to classify animals from their sounds (Zooniverse).

The AudioMoth can be loaded with simple classification algorithms to trigger logging of acoustic activities of interest. This allows only data of interest to be saved which saves on storage and work needed to pre-process the data. The AudioMoth is energy efficient and cheap, retailing at $ 94, compared to other proprietary passive acoustic monitoring devices. The Open Acoustic Devices also produces other open access audio recording and processing boards which include the HydroMoth, µMoth and AudioMoth Dev (Open-Acoustics-Devices).

### 3.4 FIELD PROGRAMMABLE GATE ARRAY (FPGA)

Field programmable gate arrays (FPGAs) are general-purpose, multilevel semiconductor devices that are built around a matrix of configurable logic blocks (CLBs) connected via programmable interconnects. The programmable interconnects allow the end users to program FPGAs to suit their needs (Brown et al., 1992; Xilinx). The ability of FPGAs to change their functionality after design gives them an edge over Application Specific Integrated Circuit (ASIC). The FPGAs also use a smaller board space and can be more energy efficient than the equivalent discrete circuit (DSL-Ltd, 2021). State of the art FPGAs possess high processing capabilities and they can be used for data processing and machine learning at the edge. The following are use cases of FPGA in data collection and processing and machine learning tasks.

#### 3.4.1 A XILINX ZYNQ BASED UNDERWATER IMAGE RECOGNITION SYSTEM

In order to study the underwater environment, researchers have developed a Xilinx ZYNQ 7000 series FPGA development board based real time image recognition system. The board has a high performance ARM CPU that provides a Linux OS for data storage, logic processing and logging. The system leverages the low power consumption, powerful computation capabilities and high flexibility characteristics of the FPGA to run a Convolutional Neural Network (CNN) for image recognition. The image recognition system is used with an autonomous underwater vehicle (AUV) for real time processing of the images collected by a camera. This was done to help the AUV respond to its surroundings in real time since communication is hindered by the limited distance that electromagnetic waves travel underwater. The power supply of the submarine cannot provide power above 100 W to the navigation system hence a microcomputer based navigation system may not be viable.

A lightweight CNN algorithm was designed to run on the FPGA for image data processing. Using parallelism and pipeline of the FPGA, parallelisation of multi-depth convolution is achieved. The input layer of the CNN was implemented on the ARM side of the board and the hidden and output layers were implemented on the FPGA side of the board. The model was trained on a powerful workstation and then loaded onto the FPGA. The model running on the FPGA was able to accurately classify captured images in a short time while consuming low power. The system was able to distinguish between driving areas like seawater and non-driving areas such as rocks and algae in a timely manner while consuming less than 10 W (Zhao et al., 2019).

#### 3.4.2 A NI MYRIO WIRELESS IMAGE TRANSMISSION FOR WILDLIFE SURVEILLANCE SYSTEM

NI myRIO is a National Instrument's microcontroller that comes with an ARM processor and an FPGA. The package has analog and digital interfaces (Princeton-University). Researchers from Malaysia and Qatar have developed a wireless image transmission wildlife surveillance system based on the NI myRIO board. The surveillance system was developed to perform real time monitoring of wildlife using images. NI myRIO board is used to collect, process (compress) and transmit images to a central computer. The images are compressed using the Discrete Cosine Transform (DCT) method to reduce the space required for their storage and resources required for transmission. LabVIEW FPGA software was used to implement the overall system. Once received on the user end, the images will be decompressed and saved from where they can be viewed for real time monitoring of wildlife (Nadzri et al., 2018).

#### 3.4.3 AN UPDUINO FPGA-ACCELERATED TIME SERIES MINING POWER IOT DEVICES

In this work, a low-cost, low-power UPDuino FPGA has been interfaced with an Arduino microcontroller to perform data processing on edge to reduce the wireless data transfer requirements. This greatly reduces the power consumed despite the addition of the FPGA to the IoT system. The UPDuino FPGA board is equipped with a Lattice iCE40 UltraPlus FPGA. In this setup, a time series similarity search was performed using the Dynamic Time Warping (DTW) algorithm. The data collection rate of the system improved by a factor of more than two while lowering the power consumption by 15% (Kang et al., 2020).

## 4 POWER CONSUMPTION OF OPEN SOURCE HARDWARE

Power consumption is one of the most important factors when designing sensors to deploy in the field. Sensors that draw less power are more desirable. The current trend in evolution of open source hardware is to improve their performance while at the same time reducing their size and power consumption (García et al., 2014). Some hardware allows for modifications in order to reduce power consumption. They include having low power and sleep modes and the ability to deactivate unused peripherals and functionalities.

In general, microcontrollers have low power requirements compared to single board computers and FPGAs. Some microcontrollers are designed to draw power in the milliwatts ranges. Microcontrollers, however, have lower processing capabilities and memory hence are limited to low computational applications. The DSAIL Water Level Data Acquisition System that is based on the mDot microcontroller has been designed to send data at intervals in between which it enters into sleep mode. In one setup, the sensor was designed to send data after every five minutes. The sensor enters sleep mode during which it draws a current of 2.2 mA and then it wakes up to send data after five minutes. The sending process takes about 3 seconds and the peak current drawn by the system during sending is 32.6 mA. This translates to an average power consumption of 12.5 mW (Kabi & wa MAINA). The AudioMoth's power consumption is as low as 80 µW during sleep mode and 17–70 mW for the most intensive tasks (Hill et al., 2018).

Single board computers and FPGAs have high processing power compared to microcontrollers but they consume more power. Single board computers like the Raspberry Pi and the Jetson Nano do not have low power or sleep modes like microcontrollers and some of the FPGAs. The DSAIL Bioacoustics System for example has an average power consumption of about 1.855 W (Kiarie & wa MAINA, 2021). Due to their high processing power, FPGAs tend to be more power hungry. The Xilinx ZYNQ based Underwater Image Recognition System described above consumes about 10 W (Nadzri et al., 2018).

## 5 CONCLUSION

Over the recent years, there has been an explosion of low-cost low-power open-source boards in the market. Newer boards with better processing capabilities are being produced year after year. Due to their improved processing power and their ability to be interfaced with sensors, the boards are being adopted for data collection and machine learning. The boards have been used to automate off-grid data collection and machine learning tasks. This paper has described how some of the open-source boards have been used to perform off-grid data collection and machine learning tasks.

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

## A   APPENDIX

The following are diagrams of some of the boards that have been described in the paper.

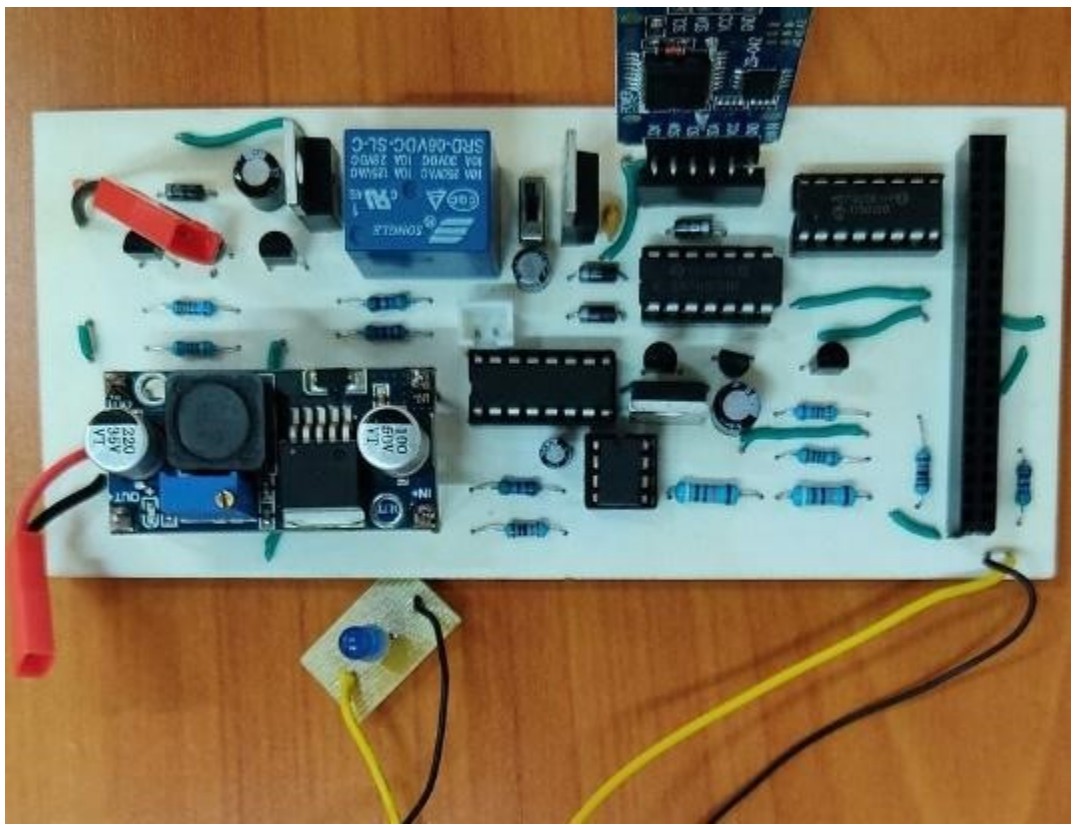

Figure 1: The DSAIL Power Management Board.

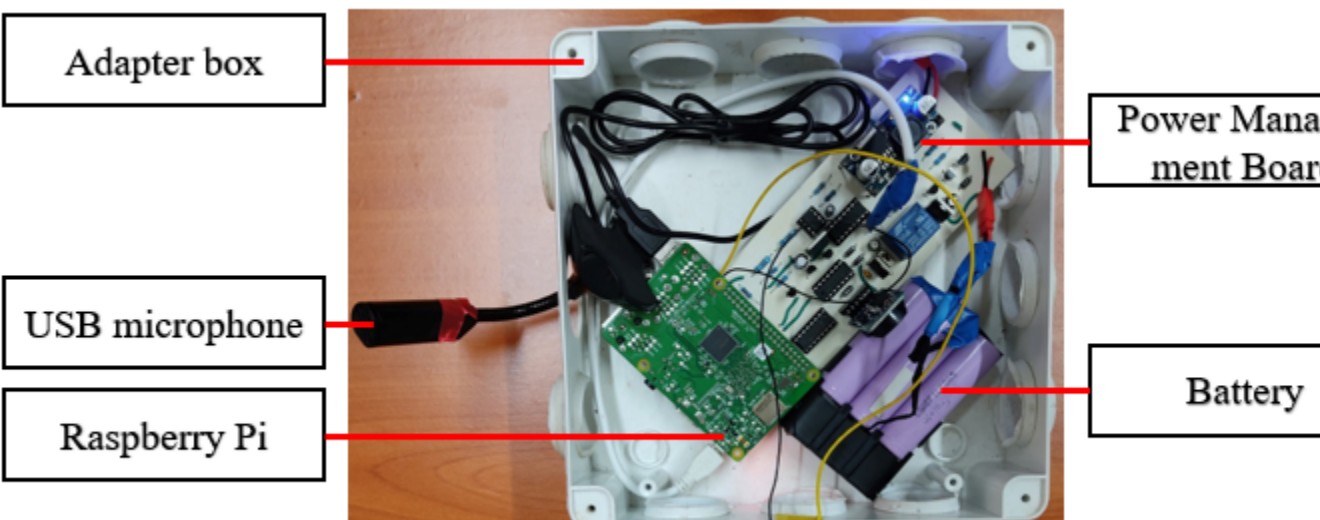

Figure 2: The DSAIL Bioacoustics System placed in an adapter box before deployment.

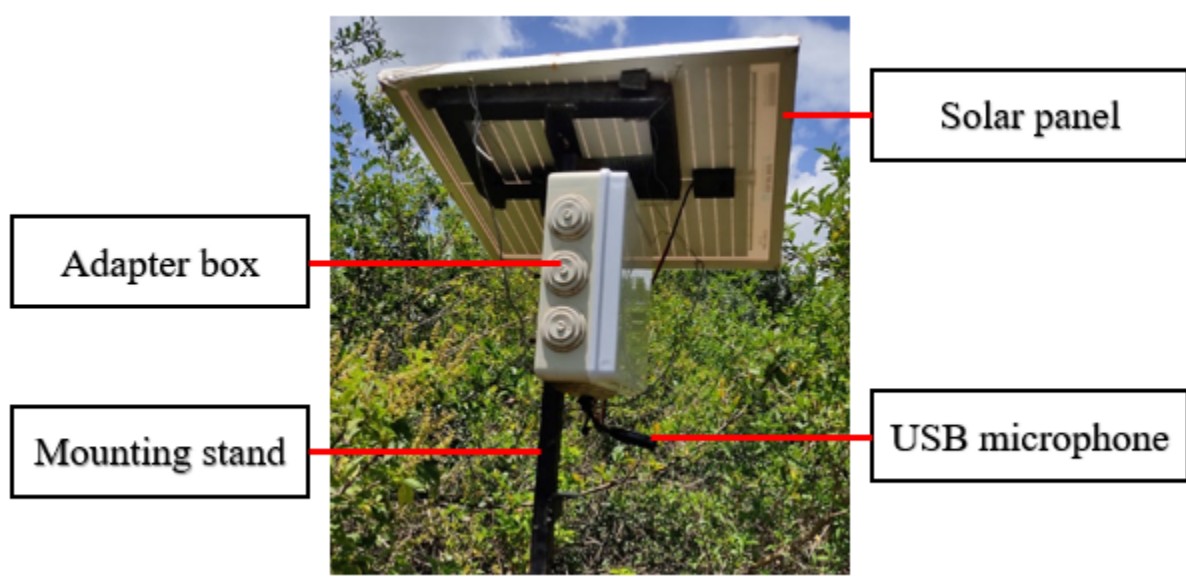

Figure 3: The DSAIL Bioacoustics System deployed at the Dedan Kimathi University of Technology Conservancy for acoustic data collection.

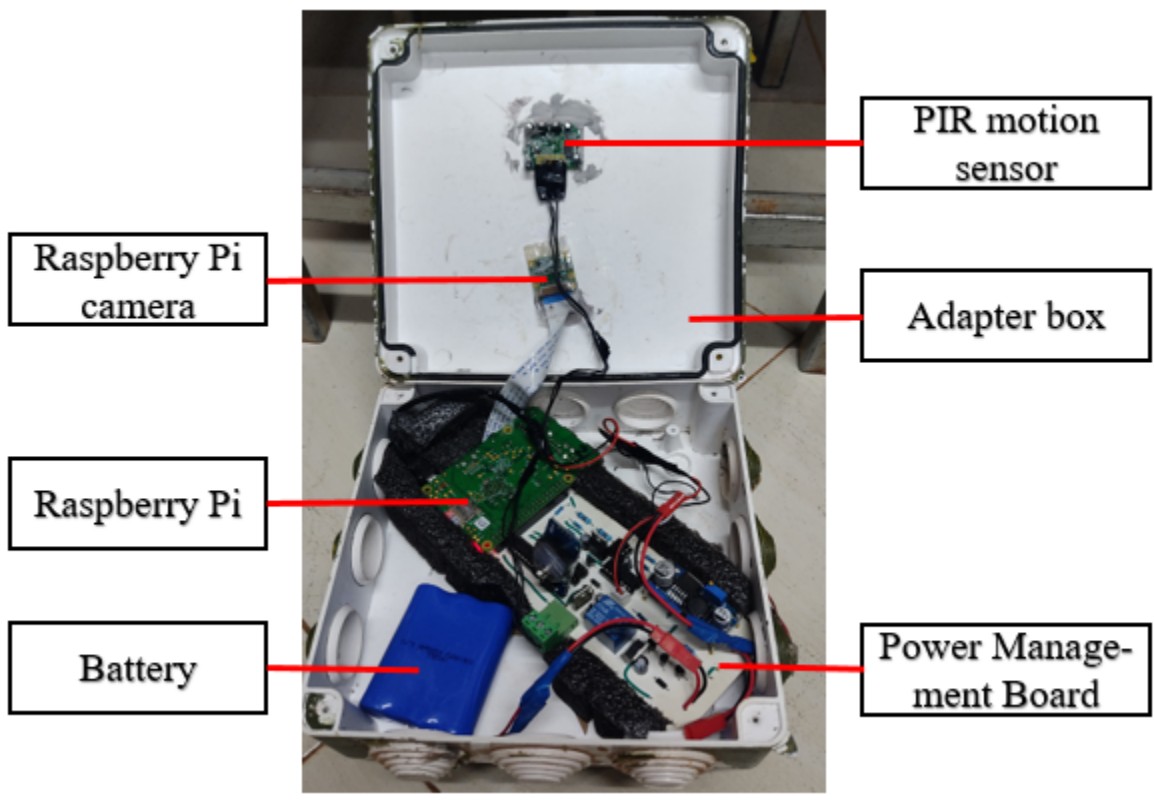

Figure 4: The DSAIL Camera Trap in an adapter box before deployment.

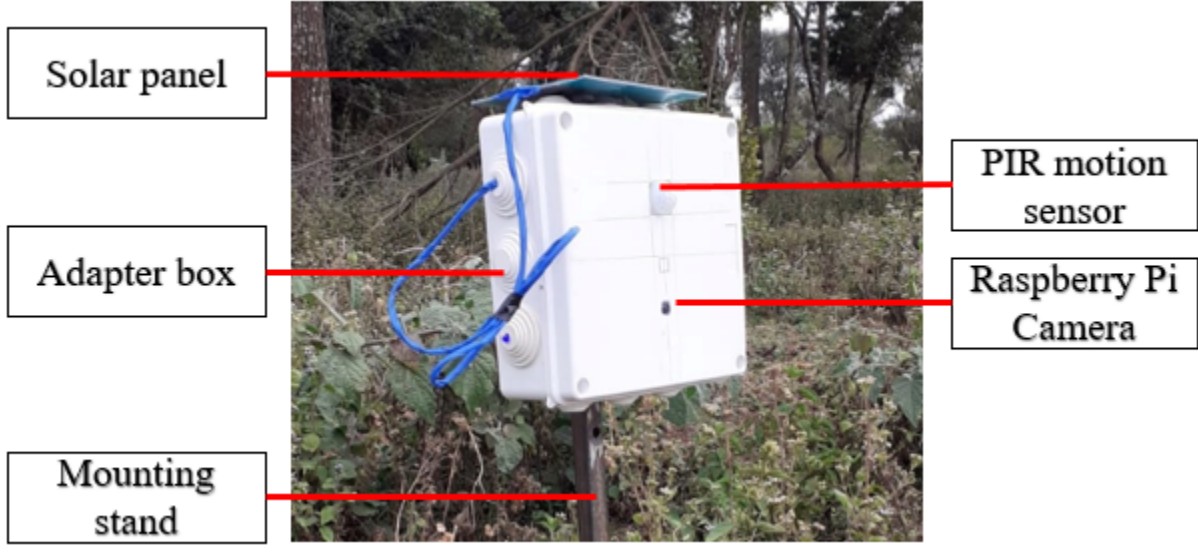

Figure 5: The DSAIL Camera Trap deployed at the Dedan Kimathi University of Technology Conservancy for data collection.

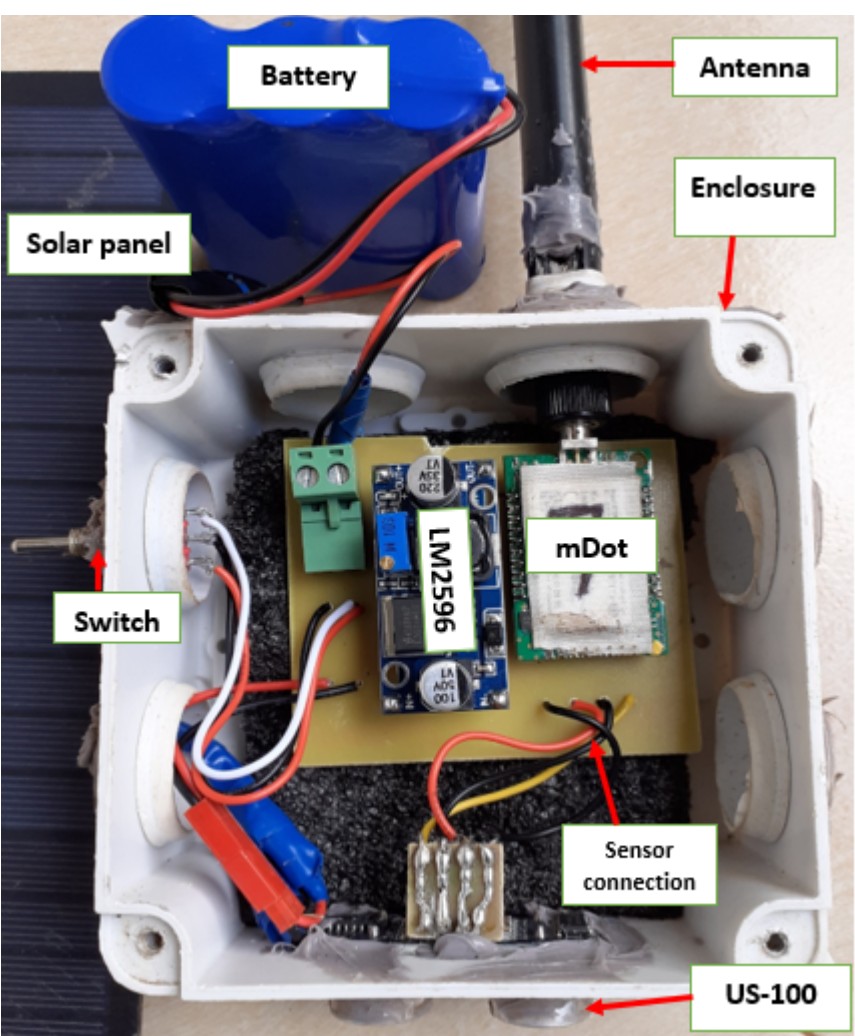

Figure 6: The DSAIL Water Level Data Acquisition System in an adapter box before deployment.

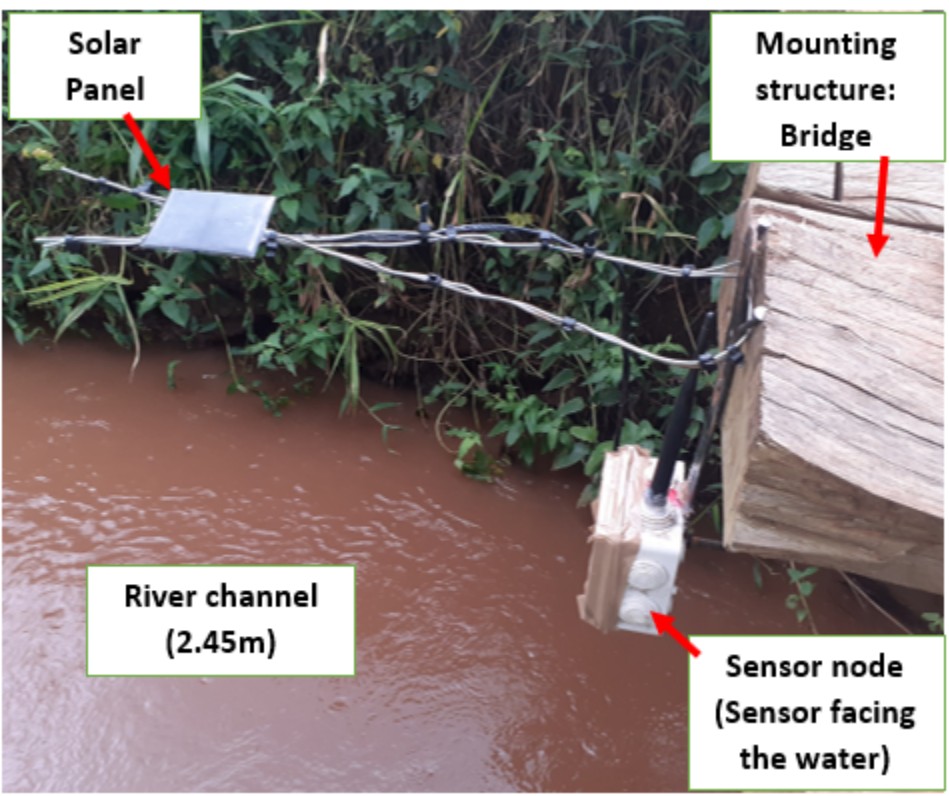

Figure 7: The DSAIL Water Level Data Acquisition System deployed at River Muringato for data collection.

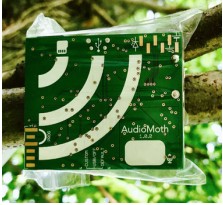

Figure 8: An AudioMoth deployed to listen for the presence of cicada species in the New Forest National Park, UK

