# OpenReview forum: "The Use of Open-Source Boards for Data Collection and Machine Learning in Remote Deployments"
_ICLR.cc/2023/Conference — Submitted to ICLR 2023_

### Official Review · Reviewer_eTp3 · 2022-10-22

**Confidence:** 5
**Correctness:** 4
**Technical Novelty And Significance:** 1
**Empirical Novelty And Significance:** Not applicable
**Recommendation:** 1

**Clarity, Quality, Novelty And Reproducibility:**

The paper is clear.
There are no experiments in the paper to evaluate quality or reproducibility.
The paper proposes no new methods to evaluate novelty.

**Strength And Weaknesses:**

Strengths
- The paper is well written, and easy to follow.

Weaknesses
- The paper is not directly relevant to machine learning or learning representations
- This is more of a survey paper, there is no novelty (new algorithms or hardware capabilities)


**Summary Of The Paper:**

The paper provides a survey of existing edge devices used for data collection in the wild. The paper explains the need for data, and how edge devices can provide this data with continuous monitoring. The devices can be powered with solar and other harvesting based techniques as well as batteries. The paper gives examples of a range of different projects that use such devices for data collection.

**Summary Of The Review:**

ICLR seeks papers that propose novel machine learning datasets, methods, and analysis. This paper is better suited at a different venue as a survey of edge devices used for ML data collection.

---

### Official Review · Reviewer_3dHd · 2022-10-24

**Confidence:** 4
**Correctness:** 4
**Technical Novelty And Significance:** 1
**Empirical Novelty And Significance:** Not applicable
**Recommendation:** 1

**Clarity, Quality, Novelty And Reproducibility:**

The paper does not provide novel insights.
The paper is clearly written.

**Strength And Weaknesses:**

Strength:
- the topic of open source hardware and resource-constrained ML is relevant
- overview over popular open source hardware
- examples for applications on this hardware

Weaknesses:
- no structured overview, instead only anecdotal examples
- no discussion or comparison of approaches
- the methods used in the applications are only described very superficially

**Summary Of The Paper:**

The paper provides a survey of data collection, data analysis and machine learning use cases on open source hardware. It first describes open access boards, their communication restrictions, and how machine learning can be performed on them. It then presents a number of use cases on different chips.



**Summary Of The Review:**

The topic of resource-aware machine learning on specialized hardware is highly relevant. The paper gives a nice anecdotal overview over applications on actual hardware. As with many survey papers, the paper has no methodological, empirical, or theoretical contribution. It does, however, also not provide a structured overview or taxonomy, no discussion of common methodology or problems, and no comparison. Therefore, it has little relevance to the community.

I suggest to build a taxonomy of open source hardware based on their capabilities and limitations and then map approaches to the hardware they are applicable to. These applications can then be exemplified by the examples already given in the paper. This would provide a structured overview over capabilities and limitations of current hardware, and the methods one can use on them.

There is a large body of work on data analysis, data mining, and machine learning on low-power devices [e.g., 1,2,3,4] and specifically machine learning at the edge [e.g., 5,6,7], as well as distributed ML on low-power devices [8]. Evaluating which approaches could be applied on which hardware would be a great contribution.

In its current form, I vote for rejection, though.


[1] Garofalakis, Minos, Daniel Keren, and Vasilis Samoladas. "Sketch-based geometric monitoring of distributed stream queries." Proceedings of the VLDB Endowment 6.10 (2013): 937-948.
[2] Piatkowski, Nico Philipp. Exponential families on resource-constrained systems. Diss. 2018.
[3] Lazerson, Arnon, Daniel Keren, and Assaf Schuster. "Lightweight monitoring of distributed streams." ACM Transactions on Database Systems (TODS) 43.2 (2018): 1-37.
[4] Lee, Sangkyun and Pölitz, Christian. Kernel Completion for Learning Consensus Support Vector Machines in Bandwidth-Limited Sensor Networks. In International Conference on Pattern Recognition Applications and Methods, 2014.
[5] Wu, Carole-Jean, et al. "Machine learning at facebook: Understanding inference at the edge." 2019 IEEE international symposium on high performance computer architecture (HPCA). IEEE, 2019.
[6] Kamp, Michael, et al. "Efficient decentralized deep learning by dynamic model averaging." Joint European conference on machine learning and knowledge discovery in databases. Springer, Cham, 2018.
[7] Murshed, MG Sarwar, et al. "Machine learning at the network edge: A survey." ACM Computing Surveys (CSUR) 54.8 (2021): 1-37.
[8] Heppe, Lukas, et al. "Resource-constrained on-device learning by dynamic averaging." Joint European Conference on Machine Learning and Knowledge Discovery in Databases. Springer, Cham, 2020.

---

### Official Review · Reviewer_sMf8 · 2022-10-24

**Confidence:** 5
**Clarity, Quality, Novelty And Reproducibility:** There is no novelty in this work as i…
**Correctness:** 3
**Technical Novelty And Significance:** 1
**Empirical Novelty And Significance:** 1
**Recommendation:** 1

**Strength And Weaknesses:**

There does not seem to be any research question in this work. Nor does there seem to be any rational to why the authors are presenting these reviews of edge devices. There is no analytical comparison of the devices.

**Summary Of The Paper:**

The paper seems to be a review of a number of edge devices which can be used for data collection.

**Summary Of The Review:**

There is no research question here nor is there any rational as to why the particular devices are compared.

---

### Official Review · Reviewer_HTpc · 2022-10-25

**Confidence:** 4
**Correctness:** 1
**Technical Novelty And Significance:** 1
**Empirical Novelty And Significance:** 1
**Recommendation:** 1

**Clarity, Quality, Novelty And Reproducibility:**

The paper is very easy to read, and the authors accurately describe what hardware was used for each of the eight applications. Furthermore, they cite the paper that introduced each system should additional details be needed.

However, the paper lacks novelty. The systems listed in this paper have already been published, and it's not clear what additional contributions this paper makes.

**Strength And Weaknesses:**

The paper has the following strengths:
* It is grammatically correct, and there is no spelling mistake.
* The paper cites the relevant related work.
* The paper summarizes a large collection of systems.

However, it's not clear what the contributions of this paper to advance the state of the art are.  The systems the paper describes have already been published before. I would have liked a direct comparison of the different systems to highlight their strength and weaknesses along various axes, such as cost per datapoint collected, reusability for other applications, size and weight to asses their relative ease of deployments, ...



**Summary Of The Paper:**

This paper describes 5 data collection and filtering systems that can be deployed in remote areas to monitor various ecosystems. These solutions have been deployed to capture the activity of various wildlife around the globe. Five of these systems are based on the raspberry pi. Three systems leverage a FPGA.



**Summary Of The Review:**

I can't recommend this paper due to a lack of novelty.

---

### Decision · Program_Chairs · 2023-01-20

**Decision:**

Reject

**Justification For Why Not Higher Score:**

Very clear reject.

**Justification For Why Not Lower Score:**

NA

**Metareview: Summary, Strengths And Weaknesses:**

All reviewers agree that this submission contains no novel research. Even as a survey paper, it lacks insightful comparisons.

**Summary Of Ac-Reviewer Meeting:**

NA